

# 1 The Detection and Attribution Model Intercomparison Project

# 2 (DAMIP v2.0) contribution to CMIP7

Nathan P. Gillett[1], Isla R. Simpson[2], Gabi Hegerl[3], Reto Knutti[4], Dann Mitchell[5], Aurélien Ribes[6], Hideo
Shiogama[7], Dáithí Stone[8], Claudia Tebaldi[9], Piotr Wolski[10], Wenxia Zhang[11], Vivek K. Arora[1]
[1]Canadian Centre for Climate Modelling and Analysis, Environment and Climate Change Canada, Victoria, Canada
[2]Climate and Global Dynamics, NSF National Center for Atmospheric Research, Boulder, CO, USA
[3]School of GeoSciences, University of Edinburgh, Edinburgh, UK
[4]Institute for Atmospheric and Climate Science, ETH Zürich, Universitätstrasse 16, 8092 Zürich, Switzerland
[5]School of Geographical Sciences, University of Bristol, Bristol, UK
[6]CNRM, Université de Toulouse, Météo-France, CNRS, Toulouse, France
[7]National Institute for Environmental Studies, Tsukuba, 305-8506, Japan
[8]National Institute of Water and Atmospheric Research, Wellington, 6021, New Zealand
[9]Joint Global Change Research Institute, Pacific Northwest National Laboratory, College Park, MD, USA
[10]University of Cape Town, Cape Town, South Africa
[11]State Key Laboratory of Numerical Modeling for Atmospheric Sciences and Geophysical Fluid Dynamics, Institute of
Atmospheric Physics, Chinese Academy of Sciences, Beijing, China
*Correspondence to*: N. P. Gillett (nathan.gillett@ec.gc.ca)





**Abstract.** The first version of the Detection and Attribution Model Intercomparison Project (DAMIP v1.0) coordinated key
simulations exploring the role of individual forcings in past, current and future climate as part of the Coupled Model
Intercomparison Project, Phase 6 (CMIP6). The simulations have been used extensively in the literature for detection and
attribution of long-term changes, constraining projections of climate change, extreme event attribution, and understanding
drivers of past and future simulated climate changes. Attribution studies using DAMIP v1.0 simulations underpinned
prominent assessments of human-induced warming in the Intergovernmental Panel on Climate Change (IPCC) Sixth
Assessment Report. Here we describe the set of DAMIP v2.0 simulations, proposed for the next phase of CMIP, CMIP7.
Detection and attribution studies rely on preindustrial control simulations and historical simulations which will be part of the
Diagnostic, Evaluation and Characterization of Klima (DECK) set of simulations for CMIP7. In addition, DAMIP v2.0
identifies three highest priority single forcing experiments for CMIP7 to be run as Fast Track simulations in support of the
Seventh Assessment Report of the IPCC - namely simulations with natural forcings only, anthropogenic well-mixed
greenhouse gases only, and anthropogenic aerosols only. Beyond this, the DAMIP v2.0 experimental design includes full
column ozone-only simulations and land-use-only simulations, such that the set of individual forcings experiments, when
considered together, represents the full set of historical forcings. While concentration driven simulations are prioritized for
attribution of past changes, emissions-driven versions of the DAMIP experiments are also proposed to support understanding
of the influence of carbon-cycle feedbacks on the simulated responses to individual forcings.





## 1 Introduction

Research on the detection and attribution of climate change aims to identify and quantify the influence of particular forcings or subsets of forcings on the climate system, with special focus on net human influence. Such research has underpinned progressively strengthening assessments on the role of human influence in driving observed climate change in successive Intergovernmental Panel on Climate Change (IPCC) reports, including the assessment in the most recent report that 'it is unequivocal that human influence has warmed the atmosphere, ocean and land' (Eyring et al., 2021). This research generally relies on climate model simulations of the response to individual forcings or subsets of external forcing, as well as observations (Eyring et al., 2021; Stott et al., 2004). In particular, such analyses generally require simulations of historical climate change including all major anthropogenic and natural influences (historical), long pre-industrial constant-forcing simulations to characterize the influence of internal variability alone (pre-industrial control), as well as simulations with subsets of forcing agents. This paper describes the coordinated set of climate model simulations designed to support detection and attribution research that are proposed under the Detection and Attribution Model Intercomparison Project v2.0 (DAMIP v2.0), part of the Coupled Model Intercomparison Project Phase 7. In CMIP7, historical and pre-industrial control simulations will be coordinated as part of the Diagnostic, Evaluation and Characterization of Klima (DECK, Dunne et al., 2024) set of simulations which all models must complete and which serve as a basis for model evaluation. Simulations with individual forcings or subsets of forcings are the focus of this paper.

## 2. Applications of DAMIP v1.0 CMIP6 simulations

DAMIP v2.0 follows from DAMIP v1.0 (Gillett et al., 2016), the coordinated set of detection and attribution simulations conducted under the Coupled Model Intercomparison Project Phase 6 (CMIP6) (Eyring et al., 2016). CMIP6 included historical simulations driven with both anthropogenic and natural forcings (historical) as well as constant pre-industrial forcing simulations (piControl). DAMIP v1.0 complemented these with historical simulations driven by subsets of the historical experiment forcings. The highest priority Tier 1 simulations consisted of historical simulations driven with natural forcings only (hist-nat), historical simulations driven with well-mixed greenhouse gas changes only (hist-GHG), and historical simulations driven with aerosol and aerosol precursor emissions changes only (hist-aer). These simulations were supplemented with lower priority Tier 2/3 simulations including simulations driven with changes in stratospheric ozone only (hist-stratO3), volcanic aerosol only (hist-volc), solar irradiance only (hist-sol) and $CO_2$-only (hist-CO2). DAMIP v1.0 extended the individual historical forcing simulations up to 2100 using SSP2-4.5 (O'Neill et al., 2016) forcings to support analysis of the contribution of the different forcings to future changes (experiments ssp245-nat, ssp245-GHG, ssp245-aer and ssp245-





stratO3). Since the publication of the original experimental design (Gillett et al., 2016), some additional experiments were
added to DAMIP v1.0, in particular: single forcing experiments with CMIP5 forcings to examine the effects of updates to the
forcings from CMIP5 to CMIP6 (Fyfe et al., 2021),  a simulation with ozone changes through the full atmospheric column
(Shiogama et al., 2023), and a set of simulations to examine the response to COVID-induced changes in emissions (Jones et
al., 2021).

Consistent with expectations, the Tier 1 simulations were carried out with the largest number of CMIP6 models, with hist-
GHG simulations from 18 models, hist-aer simulations from 17 models, and hist-nat simulations from 17 models published on
the CMIP6 data portal as of July 31st 2024 (Figure 1a). The simulations carried out with the fewest models were those added
later in the CMIP6 cycle, namely the hist-GHG-cmip5, hist-aer-cmip5, hist-nat-cmip5, ssp245-cov-GHG and ssp245-cov-aer
simulations, which were each only carried out with one model. Some modelling groups are now expanding their ensemble
sizes through the Large Ensemble Single Forcing Model Intercomparison Project (LESFMIP) (Smith et al., 2022), which is
particularly focussed on attribution of multi-annual to decadal changes in climate, including the effects of updates to forcing
datasets. While all DAMIP v1.0 experiments were referred to in at least one publication, the Tier 1 simulations (hist-GHG,
hist-aer and hist-nat) were by the far the most cited, with hist-nat referred to in 245 publications (Figure 1b). The DAMIP v1.0
simulations were also used extensively in the 6th Assessment Report of Working Group I of the IPCC, with data from these
simulations used in figures in five chapters of the report (Canadell et al., 2021; Doblas-Reyes et al., 2021; Eyring et al., 2021;
Fox-Kemper et al., 2021; Szopa et al., 2021). Here also the Tier 1 simulations were by far the most used. DAMIP v1.0
simulations were, for example, used in two attribution analyses of warming since preindustrial (Gillett et al., 2021; Ribes et
al., 2021), which were two of the three main studies used to assess the anthropogenic contribution to observed warming (Eyring
et al., 2021), which was a headline result in the Summary for Policymakers of the report (IPCC, 2021), and was also directly
quoted in the Glasgow Climate Pact (UNFCCC, 2022). Since the publication of the IPCC AR6, a selection of key climate
indicators have been updated on a yearly basis (Forster et al., 2024). This includes warming attributable to human influence,
calculated from DAMIP v1.0 data.

Beyond assessments of global temperature change, DAMIP simulations have been used to explore a wide variety of Earth
System Changes.  They  have been used to explore the role of individual forcings in modelled historical and projected future
changes in the Atlantic Meridional Overturning Circulation (AMOC) (Menary et al., 2020). While the CMIP6 ensemble mean
AMOC response to forcings was rather linear in the forcing, this is not true of all models (Simpson et al., 2023). In addition,
DAMIP simulations have been used to assess the relative roles of greenhouse gases and aerosols in historical changes in



drought frequency, duration, and intensity (Chiang et al., 2021), the relative role of anthropogenic and natural forcings in
contributing to increasing fire weather in the western United States (Zhuang et al., 2021), and the role of natural forcing,
greenhouse gases and anthropogenic aerosols in historical changes in precipitation variability (Zhang et al., 2024b). They
have been used to attribute the observed weakening of the summertime Eurasian jet stream in the historical record to
anthropogenic aerosol forcing (Dong et al., 2022) and to isolate the relative role of greenhouse gases and aerosols to Northern
Hemisphere summertime storm track trends (Kang et al., 2024). Despite this progress, we still do not adequately understand
the relative role of forced changes and internal variability in observed circulation changes. Having now observed forced signals
emerge for longer in observations and with continued improvements in process representation in Earth System Models,
together with improved estimation of external forcings, further advances in this area will likely be achieved with the use of
DAMIP v2.0 simulations of CMIP7.

As well as studies attributing long-term changes in climate, another application of DAMIP v1.0 simulations is in extreme event
attribution, which aims to characterise how the probability of a single weather or climate event was altered by specific forcings,
in particular anthropogenic forcing (Christidis et al., 2023; Herring et al., 2022; Lanet et al., 2024). This is an important use of
DAMIP data, and allows for a wider global engagement in this field than those who have the capability to run climate model
experiments tailored to the specific event under analysis. However, it also presents an additional potential challenge.
Approaches for extreme event attribution sometimes require large ensembles of climate simulations, i.e., larger than the
majority of DAMIP models will have (5-10 members), and up to (for very extreme events) the order of tens thousands of
ensemble members (Schaller et al., 2014). Therefore when DAMIP simulations are used for extreme event attribution analysis,
methods for increasing the sample size might sometimes be needed. Such methods could include using model years beyond
that of the specific event, combining different climate models and their ensembles (King, 2017), increasing the sample size
using extreme value theory (Sippel et al., 2015), or ensemble boosting (Fischer et al., 2023).

Increasingly, single forcing climate simulations are being used in hazard and impacts research. DAMIP data has been used
very effectively in going most of the way to explaining the observed impact changes, for both trend-based impact attribution,
and event-based impact attribution. In many of these studies, the authors are able to show a plausible explanation for the
climate-influenced part of the observed impact. This has happened primarily in the areas of hydrology (Li et al., 2024a), and
human health (Carlson, 2024; Chapman et al., 2022; Vicedo-Cabrera et al., 2021; Zhang et al., 2022). A challenge for impact
detection and attribution research is to effectively integrate the relevant different socio-economic factors into the  attribution
framework, but this can be particularly insightful in adaptation, loss and damage, and legal settings (James et al., 2019).




## 3 Experimental design of DAMIP v.2.0

In most respects DAMIP v2.0 follows the experimental design of DAMIP v1.0 (Table 1, Figure 2). There are two primary designs for detection and attribution experiments - namely individual forcing simulations, and all-but-one simulations, in which all forcings except the one of interest are included (Gillett et al., 2016; Smith et al., 2022). All-but-one simulations, together with historical simulations, may offer some advantages for detecting the presence of one particular forcing in the observations, in particular from a causality theory point of view (Hannart et al., 2016; Naveau et al., 2020), and they can be used together with individual forcing simulations to test additivity (Marvel et al., 2015; Shiogama et al., 2013; Simpson et al., 2023). The Large Ensemble Single Forcing Model Intercomparison Project (LESFMIP) proposed a comprehensive set of such simulations to investigate such questions (Smith et al., 2022). However, if the objective of an analysis is to characterize the response to one particular forcing, and the response to a set of forcings combined is equal to the sum of the responses to each forcing individually, then individual forcing simulations will lead to reduced sampling uncertainties, because they do not require a difference between two sets of simulations to be taken. For this reason, and for ease of comparison with previous CMIP generations, including DAMIP v1.0 experiments, DAMIP v2.0 largely follows DAMIP v1.0 in being primarily based on individual forcing simulations. That said, natural-only (hist-nat) simulations can be equivalently described and used as all-but-anthropogenic forcing simulations, and DAMIP v2.0 links with an AerChemMIP2 simulation with all forcings but aerosols (hist-piAer), which can be used to address particular questions relating to the dependence of the anthropogenic aerosol impact on the climate state and additivity of the aerosol response with the responses to other forcings (Marvel et al., 2015; Shiogama et al., 2013; Simpson et al., 2023).

Recognising the advances in modelling which are allowing a larger fraction of climate models to be run with interactive $CO_2$, the new science questions which may be addressed using interactive $CO_2$ simulations, as well as the interest in running many CMIP7 simulations with interactive $CO_2$ (Sanderson et al., 2024), DAMIP v2.0 is also proposing a set of individual forcing interactive $CO_2$ simulations (Section 3.3). As in CMIP6, such interactive $CO_2$ simulations will have distinct experiment names from their corresponding prescribed concentration simulations (because the prognostic $CO_2$ concentration will generally differ from that prescribed in the corresponding concentration-driven experiments). Interactive $CO_2$ historical simulations will sample over uncertainties in the understanding and representation of the carbon cycle within ESMs, and will allow analysis of the effects of carbon cycle feedbacks on the responses to individual forcings. However, given that direct and accurate



observations of the evolution of atmospheric $CO_2$ exist, we recommend that prescribed-$CO_2$ simulations should continue to be
used for detection and attribution studies of changes in the physical climate. For this reason, our highest-priority Fast Track
simulations are concentration-driven, as are the other simulations described in Sections 3.1 and 3.2 (the Fast Track simulations
are a subset of CMIP7 simulations to be carried out first in support of the IPCC Seventh Assessment Report (Dunne et al.,
2024)). If modelling centres have the capacity and interest to carry out both prescribed-$CO_2$ and interactive-$CO_2$ simulations,
we recommend that they carry out both, allowing the effects of carbon cycle feedbacks on the responses to individual sets of
forcings to be isolated in each model.

Like the effects of an interactive carbon cycle, atmospheric chemistry also has the potential to modify the simulated response
to individual forcings or sets of forcings. In a model with a full representation of atmospheric chemistry, methane and
fluorinated gas concentrations influence tropospheric and stratospheric ozone concentration, solar and volcanic forcings
influence stratospheric ozone concentrations, and aerosols and aerosol precursors influence ozone and methane concentrations,
among other interactions. Such interactions would make DAMIP simulations from models with and without interactive
chemistry not fully comparable. Because of such interactions, attribution of physical climate changes to changes in
concentrations of radiatively active species is fundamentally different to attribution to emissions changes of such species.
Typically detection and attribution studies of observed changes in climate attribute to concentration changes (Eyring et al.,
2021), while emissions-driven individual forcing simulations may be used for bottom-up model-based estimates of the climate
response to emissions of individual species (Szopa et al., 2021). Like DAMIP v1.0, our focus here is mainly on supporting
attribution to concentration changes, while AerChemMIP2 is proposing simulations to address attribution to emissions. While
DAMIP v1.0 did propose an experimental design for hist-GHG and hist-aer for models with interactive chemistry to try to
maintain comparability with other models, we note that this was never implemented because no modelling centres actually
carried out the DAMIP v1.0 experiments with models with interactive chemistry. To simplify the experimental design and
ensure comparability between outputs from different models, we therefore suggest that modelling groups carry out the DAMIP
v2.0 experiments using model versions without gas phase chemistry if possible. While we note that aerosol microphysics and
chemistry schemes may make simulated aerosol concentrations sensitive to simulated temperatures, winds, and possibly
greenhouse gas concentrations, and therefore different in historical and hist-aer simulations, we note that the primary sensitivity
of aerosol concentrations is to aerosol and aerosol precursor emissions, and that most modelling centres do not have the
capacity to run their models with specified aerosol concentrations, and therefore we accept such small differences as a
limitation of our experimental design.



If modelling centres plan to only submit simulations with interactive gas phase chemistry and are submitting historical
simulations including this interactive chemistry, we request that they follow the DAMIP experimental design as indicated, but
with the following modifications (see also Table 1). For hist-GHG they should specify emissions rather than concentrations of
all well-mixed greenhouse gases simulated interactively. hist-O3 simulations should not be carried out using models with
interactive gas-phase chemistry because ozone is simulated interactively in response to changes in ozone depleting substances,
methane and other species in such models, rather than prescribed directly, and the concentrations of ozone depleting substances
and methane do not change in these simulations. All other simulations should be carried out as specified.

While hist-nat simulations in models with gas phase chemistry will include changes in stratospheric ozone which may modulate
the response to solar and volcanic forcing, we expect the effects on surface climate to be small, and such output could probably
be used together with historical simulations to attribute surface climate change to anthropogenic and natural forcings. Similarly
hist-lu simulations from models with and without interactive chemistry are expected to be comparable. hist-GHG experiments
are expected to differ between models with and without gas phase chemistry, because in models with interactive chemistry,
methane emissions will increase tropospheric ozone, while halocarbon emissions will decrease stratospheric ozone. hist-aer
experiments will also differ because emissions of NOx, NMVOCs and carbon monoxide (which are aerosol precursors) will
all change tropospheric ozone and methane concentrations. Such experiments would not be directly comparable between
models with and without gas phase chemistry, but analysis of the output could inform understanding of atmospheric chemistry
feedbacks. To facilitate analysis of the results, we request that modelling centres flag output from models with interactive
chemistry with the 'f2' flag, and indicate in their metadata and documentation that the model included interactive chemistry.
We note that AerChemMIP2 also provides a dedicated framework for analysing the effects of such interactions systematically
across models with interactive chemistry.



**3.1 Historical simulations**
**3.1.1 Increased ensemble size for CMIP7 historical and Medium scenario simulations**
While our focus is on historical experiments forced with single forcings or subsets of forcings, most detection and attribution
analyses also use historical simulations with a complete set of forcings, and such analyses generally require an ensemble of
such simulations, but only a single historical simulation is required as part of the DECK (Dunne et al., 2024). Therefore we



request at least three ensemble members of the CMIP7 historical simulations, and the extension of those simulations to 2035
under the Medium scenario (intended to represent a frozen policy scenario) proposed by ScenarioMIP for CMIP7 (van Vuuren
et al., 2025). Given that actual anthropogenic forcings are expected to diverge only slightly from the Medium emissions
scenario over the first decade or so, we request that these historical simulations and other DAMIP experiments are extended
in this way. This will allow researchers to carry out attribution analyses based on contemporary data over the next decade, at
least in the absence of a major volcanic eruption, and will likely ensure an overlap with the next phase of CMIP. DAMIP v1.0
simulations were extended in a similar way from 2015 to 2020, but in hindsight this was not long enough, since at the time of
writing CMIP7 simulations are not yet available, but observations are available up to the end of 2024, well beyond the end of
the DAMIP historical simulations. Such a need is particularly apparent for regularly updated attribution analyses (Forster et
al., 2024). Modelling groups should publish the output data as CMIP7 historical simulations (1850-2021) and the Medium
scenario simulations of ScenarioMIP (2022-2035). Note that we also request an ensemble size of at least three for all other
DAMIP v2.0 historical simulations, though we encourage groups to runs larger ensembles if possible. Also note that all DAMIP
v2.0 simulations except historical-CMIP6 use forcings from the CMIP7 historical/esm-historical and Medium simulations
described by Dunne et al. (2024).

### 3.1.2 Simulations with a complete set of forcings

We are proposing a set of historical simulations driven with subsets of forcings which together sum to give the full set of
CMIP7 historical forcings. This set includes the hist-nat, hist-GHG and hist-aer experiments which are designated as FastTrack
experiments here (Dunne et al., 2024), based on the extensive use of the corresponding DAMIP v1.0 simulations in the
literature and in support of IPCC assessment reports (Figure 1b). If the climate response to these forcings is additive, then the
sum of the climate responses in this set of simulations will be equal to the response in the historical experiment. This involves
a minor adjustment to the DAMIP v1.0 experimental design in which land-use and land-cover change and tropospheric ozone
changes were not included in any of the original DAMIP v1.0 simulations. All these simulations should be run from 1850 to
2035 using the CMIP7 historical forcings, and the Medium scenario from 2022 to 2035, and we request a minimum ensemble
size of three for all historical experiments.

**hist-nat**: These natural-only simulations parallel the historical and Medium scenario simulations but instead are forced with
only solar and volcanic forcings from the historical and Medium scenario simulations, similar to the CMIP6 hist-nat
experiment. Such simulations, when compared with historical and Medium simulations, can be used for attribution of observed





changes to anthropogenic influence, as they correspond to the counterfactual world in which human influence is removed. We note that while much of the time evolution of biomass burning emissions has occurred as a result of human activity, the historical simulation includes observed year-to-year variations in biomass burning which includes a component related to natural variability, but we include that in hist-aer and request that modelling centres specify biomass burning emissions as in the piControl in this hist-nat simulation. In contrast with DAMIP v1.0, and consistent with our aim of ensuring a complete set of forcings across this simulation set and to simplify the experimental design, we are proposing that no ozone changes are specified in the hist-nat experiment, and that all ozone changes are included in the hist-O3 experiment.

**hist-GHG**: These greenhouse-gas-only simulations parallel the historical and Medium simulations but are forced by well-mixed greenhouse gas (carbon dioxide, methane, nitrous oxide and fluorinated gas) changes only from the historical and Medium scenario simulations, similarly to the CMIP6 hist-GHG experiment. Both stratospheric and tropospheric ozone should be held constant at piControl levels. Greenhouse gas changes are the dominant anthropogenic forcing, and these simulations will allow the response to this forcing to be quantified. Moreover, historical/Medium, hist-nat and hist-GHG will together allow the attribution of observed climate change to natural, greenhouse gas and other anthropogenic forcings (e.g. Gillett et al., 2021; Ribes et al., 2021).

**hist-aer**: These historical aerosol-only simulations parallel the historical/Medium simulations but are forced by changes in aerosol and aerosol precursor emissions changes only, as in historical and the Medium scenario. This includes changes in sulphur dioxide, black carbon, organic carbon, ammonia, NOx and VOCs, from biomass burning, industrial emissions and other sources.

**hist-lu:** These simulations parallel the historical and Medium simulations but are forced with prescribed land-use and land cover changes only from the historical and Medium scenario simulations, with all other forcings held constant at 1850 values. No such experiments were included in DAMIP v1.0, although they have since been proposed in LESFMIP (Smith et al., 2022), and historical experiments without land use change (hist-noLu) were included in CMIP6 in the Land Use Model Intercomparison Project (LUMIP, Lawrence et al., 2016). These experiments were used to investigate the effects of land use change on surface temperature and other variables in models (Luo et al., 2024; Zhang et al., 2024a), and to diagnose land use change emissions (e.g. Liddicoat et al., 2021). Note that hist-noLu is also a proposed LUMIP experiment for CMIP7, and hence hist-lu and hist-noLu could be used together to evaluate the additivity of the land-use change response and the response to other forcings. These experiments could also for example be used together to compare and contrast simulated historical land-use change emissions with atmospheric $CO_2$ held constant, and in the presence of $CO_2$ fertilisation with $CO_2$ changing through the historical period.





**hist-O3**: These simulations parallel the historical/Medium simulations but are forced by changes in ozone concentrations only from the historical and Medium scenario simulations. They will allow characterization of the response to combined tropospheric and stratospheric ozone changes, which have played an important role in driving circulation changes in the high-latitudes of the Southern Hemisphere and temperature changes in the stratosphere, as well as attribution studies of the response to ozone change (e.g. Gillett et al., 2013; Morgenstern, 2021). Since it is increasingly understood that ozone depleting substances influence ozone concentrations in the troposphere (e.g. Hassler et al., 2022; Li et al., 2024b), in CMIP7 we are requesting simulations forced by changes in ozone concentrations over the whole atmospheric column, as opposed to the stratospheric ozone changes only simulations proposed in CMIP6 (Gillett et al., 2016) (although experiments with ozone changes over the whole atmosphere were later proposed and carried out (Liu et al., 2022; Shiogama et al., 2023; Smith et al., 2022)). Moreover, this choice simplifies the experiment design, and avoids difficulties associated with specifying stratospheric ozone changes only, given that tropopause height can differ across models.

### 3.1.3 Other historical simulations

**hist-volc**: The hist-volc simulations parallel the hist-nat simulations except that the hist-volc simulations are driven by stratospheric aerosol changes only. The hist-volc experiments will allow the characterisation of and attribution to volcanic influence, separate from the poorly constrained response to variations in solar forcing. Such analysis can improve our understanding of the response to future volcanic eruptions.

**historical-CMIP6:** These are identical to CMIP7 historical experiments, but instead of using CMIP7 forcings, they use CMIP6 historical forcings to 2014 and ScenarioMIP SSP2-4.5 forcings from 2015 to 2035. CMIP simulations of past and future climate changes are key to attribution as well as projection of climate change, for example in IPCC assessments (Eyring et al., 2021; Lee et al., 2021), and it is important to be able to understand differences in results based on successive CMIP generations. Comparing historical CMIP6 and CMIP7 simulations is complicated by the fact that both the models and the forcings are different between the two CMIP generations, making it difficult to separate the influence of updates to the forcings versus changes to the models. By carrying out simulations with a subset of CMIP7 models using CMIP6 forcings, it will be possible to separate the influence of updates to the forcings. A similar approach was used to isolate the influence of updates to forcings between CMIP5 and CMIP6 and the impact of forcing changes has been shown to be an important contributor to differences between CMIP6 and CMIP5 simulations (Fyfe et al., 2021; Holland et al., 2024).



## 3.2 Future simulations driven with subsets of forcings

While ScenarioMIP simulations including scenarios of future changes in greenhouse gases, aerosols, ozone and land use change allow the simulation of possible future climate evolution, they do not directly allow the effects of particular forcings, or sets of forcings, to be isolated. Simulations of future climate change with subsets of forcings can help us understand the drivers of future climate change. Moreover, the Allen et al. (2000) and Stott and Kettleborough (2002) approach to constraining climate projections may be used to separately scale the future response to well-mixed greenhouse gases and to aerosols, based on regressions over the historical period, since models may over- or under-estimate the climate response to greenhouse gases and aerosols by different factors. This kind of analysis requires future scenario simulations with subsets of forcings. DAMIP v1.0 included ssp245-nat, ssp245-GHG, ssp245-aer and ssp245-stratO3 simulations to address such needs (Gillett et al., 2016). Similarly, DAMIP v2.0 also includes future simulations with subsets of forcings following the Medium scenario from the ScenarioMIP design for CMIP7 (van Vuuren et al., 2025), which approximately corresponds to a continuation of current climate policies.

**Medium-GHG**: This is an extension of the hist-GHG simulations to 2100 using the Medium scenario well-mixed greenhouse gas concentrations, with other forcings kept at pre-industrial values.

**Medium-aer**: This is an extension of the hist-aer simulations to 2100 following the Medium scenario aerosol concentrations/emissions, with other forcings kept at pre-industrial levels. These simulations will also allow a more robust characterisation of the response to future aerosol changes, without conflating these changes with the responses to ozone and land-use changes, as would occur if the aerosol response were estimated from a difference between the Medium scenario and Medium-GHG.

**Medium-O3**: These simulations are extensions of the hist-O3 simulations to 2100 following the ozone concentrations specified for the Medium scenario. Stratospheric ozone is projected to recover following the successful implementation of the Montreal Protocol and its amendments (e.g. Hassler et al., 2022). These simulations will facilitate a robust multi-model assessment of the climate effects of this recovery on Southern Hemisphere climate and stratospheric temperature.

## 3.3 Interactive $CO_2$ experiments

Detection and attribution studies have typically attributed changes in the physical climate system to changes in concentrations of greenhouse gases and other species by regressing observed changes onto the simulated response to changes in concentrations of greenhouse gases and other species (e.g. Eyring et al., 2021), but an alternative bottom-up approach uses models directly to





simulate the responses to changes in emissions of particular greenhouse gases or other species (e.g. Forster et al., 2021; Szopa
et al., 2021). Such approaches can quantify the respective contributions of emissions of various chemical species, such as $CO_2$,
methane, $SO_2$, and others. These approaches can account for feedbacks on the response to emissions of particular chemical
species, based on their representation in models. IPCC (2021) contrasted observationally-constrained attributable warming in
response to changes in concentration of greenhouse gases among other factors, with estimates of the warming attributable to
emissions of $CO_2$, methane and other species. These estimates included the simulated effects of tropospheric chemistry (Szopa
et al., 2021), which strongly influence the responses to some forcings, but they omitted the effects of carbon-climate feedbacks
which are also expected to influence the responses to such forcings (Szopa et al., 2021). In part to address this knowledge gap,
we are proposing interactive-$CO_2$ versions of all the experiments described in Sections 3.1 and 3.2, with the exception of
historical-CMIP6 (where our focus is on understanding differences in the concentration-driven historical experiments between
CMIP6 and CMIP7). Such experiments should be carried out using Earth System Models (ESMs) with interactive carbon
cycles, which can simulate atmospheric concentrations of $CO_2$ based on prescribed emissions. As noted above, we recommend
that modelling centres carrying out these interactive $CO_2$ experiments also carry out the corresponding prescribed $CO_2$
experiments, allowing the effects of carbon-climate feedbacks on the response to each set of forcings to be quantified. These
simulations should be run from 1850 to 2035, using the esm-historical $CO_2$ emissions and other forcings for the period 1850-
2021, and the Medium scenario $CO_2$ emissions and other forcings for the period 2022 to 2035.

**esm-hist-nat**: These interactive-$CO_2$ simulations parallel esm-historical simulations, but with only solar and volcanic forcings
varying, and all other forcings held fixed. Such simulations can be used for example to evaluate the effects of carbon-climate
feedbacks on the response to volcanoes (Kandlbauer et al., 2013; Rothenberg et al., 2012), or as a true counterfactual simulation
reflecting the $CO_2$ evolution in the absence of anthropogenic influence.

**esm-hist-GHG**: These interactive-$CO_2$ simulations parallel the esm-historical simulations, but include only fossil $CO_2$
emissions and changes in the concentration of other well-mixed greenhouse gases (methane, nitrous oxide and fluorinated
gas). No changes in land-use or emissions associated with land-use change should be prescribed.

**esm-hist-aer**: These interactive-$CO_2$ simulations parallel the esm-historical simulations, but include only changes in aerosol
and aerosol precursor emissions. Changes in aerosols can perturb the carbon cycle not only through changes in the climate,
but also through deposition of nutrients such as nitrogen, sulphur, iron and phosphorous, and through changes in solar



irradiance and diffuse radiation at the surface (Szopa et al., 2021), and the response will depend on the representation of these
mechanisms in each ESM.

**esm-lu**: These interactive-$CO_2$ simulations parallel the esm-historical simulations but are forced with prescribed land-use and
land cover changes only, with all other forcings held constant at 1850 values. Such experiments will include the simulation of
$CO_2$ emissions based on prescribed land-use change, and will support the calculation of the net climate influence of land-use
change (biogeophysical and carbon dioxide effects) (e.g. Intergovernmental Panel On Climate Change, 2022).

**esm-hist-O3:** These interactive-$CO_2$ simulations parallel the esm-historical simulations, but include only prescribed changes
in tropospheric and stratospheric ozone concentration. Tropospheric ozone increases reduce terrestrial plant growth influencing
land carbon uptake (Szopa et al., 2021). Early studies found a substantial role for stratospheric ozone in driving changes in the
Southern Ocean carbon sink (Le Quéré et al., 2007), but more recent assessments find a weaker role (Garny, H et al., 2022).
These simulations will support the investigation of the effects of such processes on atmospheric $CO_2$ concentration.

**esm-hist-volc**: These interactive-$CO_2$ simulations parallel the esm-historical simulations, but are driven with stratospheric
aerosol changes only.

**esm-Medium-GHG**: Thes interactive $CO_2$ simulations are extensions of the esm-hist-GHG simulations, but with $CO_2$
emissions and other well-mixed greenhouse gas concentrations following the Medium scenario.

**esm-Medium-aer:** Thes interactive $CO_2$ simulations are extensions of the esm-hist-aer simulations, but with aerosol and
aerosol precursor emissions following the Medium scenario.

**esm-Medium-O3:** These interactive $CO_2$ simulations are extensions of the esm-hist-O3 simulations, but with tropospheric
and stratospheric ozone concentration following the Medium scenario.

Figure 3 shows global mean surface $CO_2$ concentration and temperature in a subset of such simulations carried out with
CanESM5.0 (Swart et al., 2019), compared with corresponding prescribed concentration simulations for reference. The
simulated $CO_2$ in the esm-historical simulations of this model is relatively close to that observed and that specified in the
historical and hist-GHG simulations (compare the solid and dashed black lines in Figure 3a). As expected, esm-hist-GHG





shows lower simulated $CO_2$ concentrations than esm-historical, because it includes only emissions of fossil $CO_2$ and omits
land-use change emissions of $CO_2$. Because of  this, esm-hist-GHG exhibits less warming than hist-GHG (Figure 3b). By
contrast esm-hist-lu shows an increase in atmospheric $CO_2$ due to the interactively simulated effects of land-use change, and
hence it is warmer than hist-lu, in which constant preindustrial $CO_2$ is specified (Figure 3b). In this model, atmospheric $CO_2$
increases slightly in esm-hist-aer. While the ocean takes up carbon in this simulation in response to the simulated cooling, this
is more than compensated for by the land giving up carbon, likely due to reduced photosynthesis in response to the reduced
solar irradiance. However, this model lacks a representation of the vegetation response to the change in diffuse irradiance
associated with the increase in atmospheric aerosols (Szopa et al., 2021), and therefore this response might be different in other
models. Because of this increase in atmospheric $CO_2$, the cooling in response to aerosols in esm-hist-aer is slightly weaker
than in hist-aer, but overall these two experiments exhibit a comparable global surface temperature response in this model.

**3.4 Updated forcing simulations**
As noted in Section 3.1, for the purposes of detection and attribution analyses, we recommend extending CMIP7 historical
simulations from 2022 to 2035 with the ScenarioMIP Medium scenario, and similarly to extend DAMIP single-forcing
simulations from 2022 to 2035 with the individual forcings prescribed for this scenario. However, observed concentrations
and emissions of forcing agents will at some point evolve differently from those specified in this scenario. While forcings have
evolved broadly consistently with the SSP2-4.5 scenario used to extend DAMIP v1.0 simulations since 2015 (Matthews and
Wynes, 2022), this has not been completely true. For example, recent aerosol emissions from China have declined more
strongly than those specified in the SSP2-4.5 scenario (Wang et al., 2021), a decline in marine sulphur emissions resulting
from new International Maritime Organisation regulations was not included in the SSP2-4.5 simulations (Watson-Parris et al.,
2022, 2024), the COVID-19 pandemic changed emissions temporarily in ways which weren't included in the SSP2-4.5
scenario (e.g. Szopa et al., 2021), and the Hunga Tonga eruption in 2021 injected sulphur dioxide and water vapour into the
stratosphere, influencing the climate (e.g. Jenkins et al., 2023). In the past such differences between forcings used to drive
climate models simulations, and those which actually have transpired have been cited as reasons for differences between
simulated and observed warming trends in the early 21st century (e.g. Eyring et al., 2021; Santer et al., 2014), and in the early
2020s (Rantanen and Laaksonen, 2024).  For these reasons, LESFMIP proposed a set of individual forcing simulations using
regularly updated forcing estimates, as part of CMIP6 (Smith et al., 2022), though at the time of writing such simulations have
not yet been carried out and the protocols and process are not in place for the regular update of forcings which would support
such experiments. If this could be achieved in the future, however, such simulations could be used to understand the
contribution of updates to particular forcing estimates in contributing to observed climate trends, particularly on interannual



to decadal timescales. Given the uncertainty in whether future forcings over the next decade or so will diverge strongly from
those specified in the Medium scenario, and if and when updated forcings datasets would be made available to the modelling
community, we are not proposing additional named experiments here. However, we do note that the Earth System Grid
Federation (ESGF) naming convention includes a forcing index which can be used to label different forcing variants for a
given experiment. We recommend that this index is used to publish simulations with updated forcing variants if and when
modelling centres perform them. For example, if the original version of hist-nat, using the Medium scenario forcings from
2022 to 2035 were published with the 'f1' forcing index, and were a major volcanic eruption to occur in 2027, and an updated
stratospheric aerosol forcing datasets be published that year, a new version of the simulation using the updated forcings could
be published with the 'f2027' forcing index. Such a simulation could cover only part of the time period covered by the original
simulation (e.g. 2022-2035 only), as long as it is labelled appropriately. We suggest labelling such simulations with major
updates to forcings with the year which historical forcings were extended to e.g. 'f2027'. Any such effort would require
additional community coordination and documentation if and when new forcing datasets were published, and would be most
valuable if carried out in conjunction with updated all-forcing scenario simulations as part of ScenarioMIP or DCPP.

**4. Synergies with other MIPs**

In CMIP6, several DAMIP v1.0 experiments were coordinated with other MIPs in order to maximise synergies and support
research to address a wide range of scientific questions (Gillett et al., 2016). For example, DAMIP v1.0 and the Radiative
Forcing MIP (RFMIP) (Pincus et al., 2016) proposed coordinated coupled and prescribed sea surface temperature simulations
with natural-only and well-mixed GHG-only forcings, in order to estimate corresponding effective radiative forcings. For
CMIP7, we have again coordinated with a broad range of MIPs to maximise scientific synergies.

As discussed above, all DAMIP v2.0 simulations extend beyond the end of the historical simulation in 2021, and therefore we
are recommending extending these simulations with the Medium scenario from ScenarioMIP. Our simulations which are
extended to 2100 are also coordinated with ScenarioMIP to support understanding of the roles of individual forcings in driving
future climate change in the Medium scenario. Similarly, we are coordinating our scenario choice with DCPP, such that both
DAMIP simulations and DCPP simulations will use the same scenario, allowing the contributions of individual forcings to
near-future climate change to be calculated and compared with initialised predictions under the same consistent set of forcings.

The AerChemMIP2 Fast Track hist-piAer experiments parallel the historical and Medium scenario simulations, except that
aerosols are kept fixed at pre-industrial levels. Together with the hist-aer and historical experiments, these experiments will





allow analysis of the extent to which the climate response to aerosols is additive with the response to other forcings. For this
reason we encourage modelling centres participating in DAMIP to also carry out the hist-piAer simulations.

The RFMIP experiment piClim-histaer (which is an atmosphere-only simulation with fixed preindustrial SSTs and sea ice and
transient aerosols) parallels hist-aer, except that it has fixed SSTs. This experiment can be used to diagnose the evolution of
the effective radiative forcing of aerosols, while the DAMIP hist-aer experiment can be used to diagnose the climate response
to that forcing.

**5. Variables requested by DAMIP**

DAMIP has provided input to the process that is defining the harmonized data request for DECK and FastTrack experiments
through the proposal of the "Detection and Attribution" opportunity.  This opportunity contains basic variables that are needed
for quantifying how the mean climate and its variability are changing over time, for understanding the mechanisms involved,
and for comparing with the observational record with a view to detecting and attributing climate change signals.  Many of
these variables overlap with the proposed baseline variable set for Earth System Modeling (Juckes et al., 2024).   These
proposed variables include fields for assessing the different forcings that the climate system is experiencing, fields for assessing
global mean temperature, hydrological, sea level, and circulation changes in both the atmosphere and ocean, as well as top of
atmosphere and surface fluxes for diagnosing energy balances and fields for understanding the role of clouds in the climate
system. With the move towards a greater role for emissions-driven simulations, we also request variables that would help us
understand the origins of differences in $CO_2$ concentration among simulations when run in emissions-driven mode. A variety
of daily fields are also requested to understand the time evolution of extremes and fields that can be used to diagnose synoptic
features and daily surface fluxes that can be used to understand the drivers of those extremes.  Finally, DAMIP requests zonal
mean or 3D atmospheric temperature with a high vertical resolution extending deep into the stratosphere to allow the upper
levels to be examined with sufficient resolution to be compared with Stratospheric Sounding Unit (SSU) observations  (e.g.
Mitchell, 2016; Santer et al., 2023). Stratospheric temperature trends are an important part of the climate change signal, and
are also particularly insightful for assessing ozone recovery, volcanic aerosol, and solar signals  (Mitchell, 2016; Santer et al.,

475  2023).


We propose the "Detection and Attribution" variable groups be output for all DAMIP experiments and related experiments
from the DECK and other MIPs shown in Figure 2. We also suggest that these be output for the pre-industrial control



experiments to allow the uncertainties due to internal variability alone to be quantified as well as the AMIP experiments which
can be used to explore the role of observed historical trends in SSTs in the evolution of the climate system and in potential
differences between coupled models and observations. Since most of the variables proposed as part of the Detection and
Attribution opportunity are basic fields needed to characterize the behavior of the climate system, we also suggest that these
variables be output for the more idealized experiments that are part of DECK and Fast Track that can be used to understand
the behavior of models and explore the direct effects of radiative forcings and the indirect effects of SST warming on the
climate system. These include: 1pctCO2, 1pctCO2-bgc, 1pctCO2-rad, abrupt-4xCO2, piClim-anthro, hist-piSLCF, amip-p4k,
and amip-piForcing. We also suggest that these variables are saved for the DCPP initialized predictions from 2025-2036 to
compare simulated near term trends in these predictions with those in free-running coupled simulations, allowing for attribution
of forthcoming trends in the climate system to external forcings as well as aspects of variability and change that are imparted
to the models through the initial conditions in this particular experiment.

**6. Summary**
This paper describes the Detection and Attribution Model Intercomparison Project (DAMIP v2.0) which will form part of
CMIP7 (Dunne et al., 2024) and is intended to underpin detection and attribution analysis informing the IPCC Seventh
Assessment Report. DAMIP v1.0 simulations in CMIP6 were extensively used in the fields of attribution of climate trends,
extreme event attribution and attribution of climate impacts, and they also underpinned the assessment of human-induced
global warming in the IPCC AR6 (IPCC, 2021), which was reported in the Glasgow Climate Pact (UNFCCC, 2022).

DAMIP v2.0 again proposes hist-nat, hist-GHG and hist-aer simulations as high priority Fast Track simulations for CMIP7.
These simulations were the most heavily used in CMIP6. We are proposing that they are extended to 2035 using the Medium
scenario from ScenarioMIP. However, we also propose a set of historical simulations with forcings which together from the
complete set of historical forcings (hist-GHG, hist-aer, hist-nat, hist-lu and hist-O3), allowing additivity to be easily tested,
and ensuring all forcings are covered. Given that stratospheric and tropospheric ozone changes interact with each other, and
to simplify the experimental design, we propose hist-O3 simulations which have prescribed ozone changes in the stratosphere
and troposphere, rather than just in the stratosphere as was the case for CMIP6. To evaluate the effects of individual forcings
on future climate evolution, we are proposing extension of key simulations to 2100 under the Medium emissions scenario.
Finally, for CMIP7 we are proposing a new set of interactive-$CO_2$ simulations, which will among other things allow the net
effect of land-use change on climate to be evaluated, and allow the effects of carbon-climate feedbacks on the simulated
response to individual forcings to be evaluated.




**Website of DAMIP:** Updated details on the project and its progress will be available at https://wcrp-cmip.org/mips/damip/.

**Code Availability:** The full CanESM5 source code, used to run the simulation shown in Figure 3, is publicly available
at https://gitlab.com/cccma/canesm (last access: 30 January 2025). The version of the code which can be used to produce all
the simulations described in this paper, is tagged as v5.0.3 and has the associated
DOI: https://doi.org/10.5281/zenodo.3251113 (Swart et al., 2019).

**Data Availability:** The model output from the DAMIP simulations described in this paper will be distributed through the
Earth System Grid Federation (ESGF) with digital object identifiers (DOIs) assigned. The model output will be freely
accessible through data portals after registration.

**Author Contributions:** As members of the Scientific Steering Committee of  DAMIP v2.0, NPG, IRS, GH, RK, DM, AR,
HS, DS, CT, PW and WZ together developed the experimental design for DAMIP v2.0. NPG prepared Figures 1 and 3, and
IRS prepared Figure 2. VKA carried out the interactive $CO_2$ simulations shown in Figure 3. NPG led the writing of the
manuscript, and all authors contributed to editing and revising the text.

**Competing Interests:** The authors declare that they have no conflict of interest.

**Acknowledgments**: CT  was supported by the Office of Science, U.S. Department of Energy Biological and Environmental
Research as part of the Water Cycle and Climate Extremes Modeling (WACCEM) project funded by the Regional and
Global Model Analysis program area. Pacific Northwest National Laboratory is operated by Battelle for the U.S. Department
of Energy under Contract DE-AC05-76RL01830. DS was supported by the Ministry of Business, Innovation and
Employment (MBIE), Aotearoa New Zealand, through the Whakahura Endeavour Programme.

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



**Table 1** DAMIP v2.0 simulations.

| Name | Description (forcing agents perturbed) | Interactive CO$_2$ equivalent simulation name (modifications) | Modifications for coupled chemistry models | Start year | End year | Minimum ensemble size |
|---|---|---|---|---|---|---|
| historical and Medium scenario | Enlarging ensemble size of historical (1850-2021) and ScenarioMIP Medium scenario (2022-2035) to an ensemble size of at least 3. Forcings: WMGHGs, BC, OC, SO$_2$, SO$_4$, NO$_x$ , NH$_3$, CO, NMVOC, nitrogen deposition, ozone, stratospheric aerosols, solar irradiance, land use) | esm-historical and esm-Medium (prescribe fossil CO$_2$ emissions instead of concentrations) | | 1850 | 2035 | 3 |
| hist-nat | Natural-only historical simulations (solar irradiance, stratospheric aerosol) | esm-hist-nat | | 1850 | 2035 | 3 |
| hist-GHG | Well-mixed greenhouse gas only historical simulations (WMGHGs) | esm-hist-GHG (prescribe fossil CO$_2$ emissions instead of CO$_2$ concentrations) | Prescribe emissions instead of concentrations of those WMGHGs simulated interactively | 1850 | 2035 | 3 |
| hist-aer | Anthropogenic aerosol-only historical simulations (BC, OC, SO$_2$, SO$_4$, NO$_x$ , NH$_3$, CO, NMVOC) | esm-hist-aer | | 1850 | 2035 | 3 |
| hist-O3 | Ozone-only historical simulations (ozone concentration) | esm-hist-O3 | NA | 1850 | 2035 | 3 |





| hist-lu | Land-use change only historical simulation | esm-hist-lu | | 1850 | 2035 | 3 |
|---|---|---|---|---|---|---|
| historical-CMIP6 | Historical simulation with forcings from CMIP6 historical and SSP2-4.5 (WMGHGs, BC, OC, $SO_2$, $SO_4$, $NO_x$ , $NH_3$, CO, NMVOC, nitrogen deposition, ozone, stratospheric aerosols, solar irradiance, land use) | NA | NA | 1850 | 2035 | 3 |
| hist-volc | Volcanic aerosol only historical simulation (stratospheric aerosols) | esm-hist-volc | | 1850 | 2035 | 3 |
| Medium-GHG | Extension of at least one hist-GHG experiment to 2100 using the Medium scenario. | esm-Medium-ghg (prescribe fossil $CO_2$ emissions instead of $CO_2$ concentrations) | Prescribe emissions instead of concentrations of those WMGHGs simulated interactively | 2036 | 2100 | 1 |
| Medium-aer | Extension of at least one hist-aer experiment to 2100 using the Medium scenario. | esm-Medium-aer | | 2036 | 2100 | 1 |
| Medium-O3 | Extension of at least one hist-O3 experiment to 2100 using the Medium scenario. | esm-Medium-O3 | NA | 2036 | 2100 | 1 |


(a)

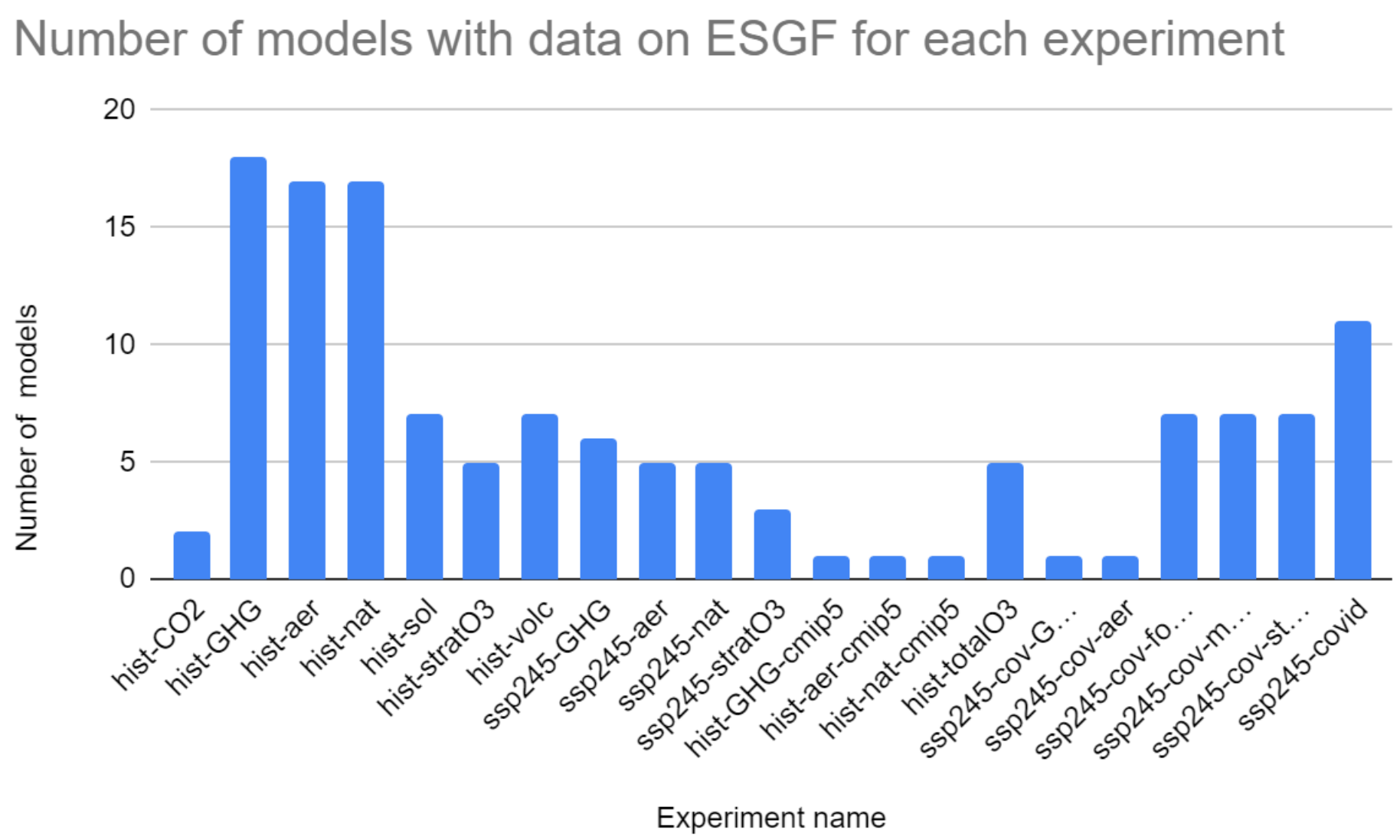

(b)

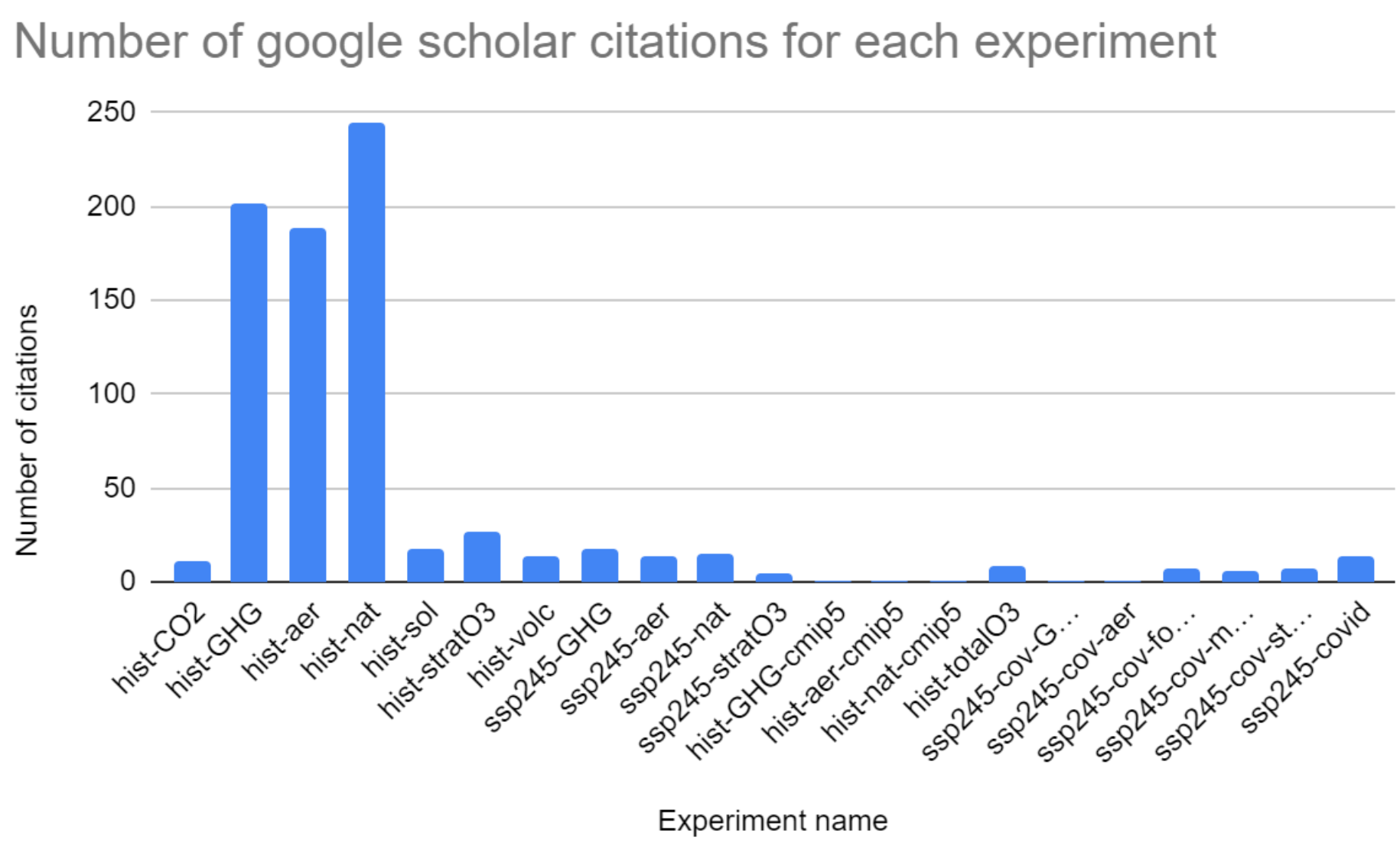

**Figure 1.** Bar graphs showing (a) the number of models with data published on ESGF for each DAMIP v1.0 experiment, and (b) the number of Google Scholar citations for each DAMIP v1.0 experiment (based on a search for 'CMIP6' and each experiment name), as of July 31st 2024.





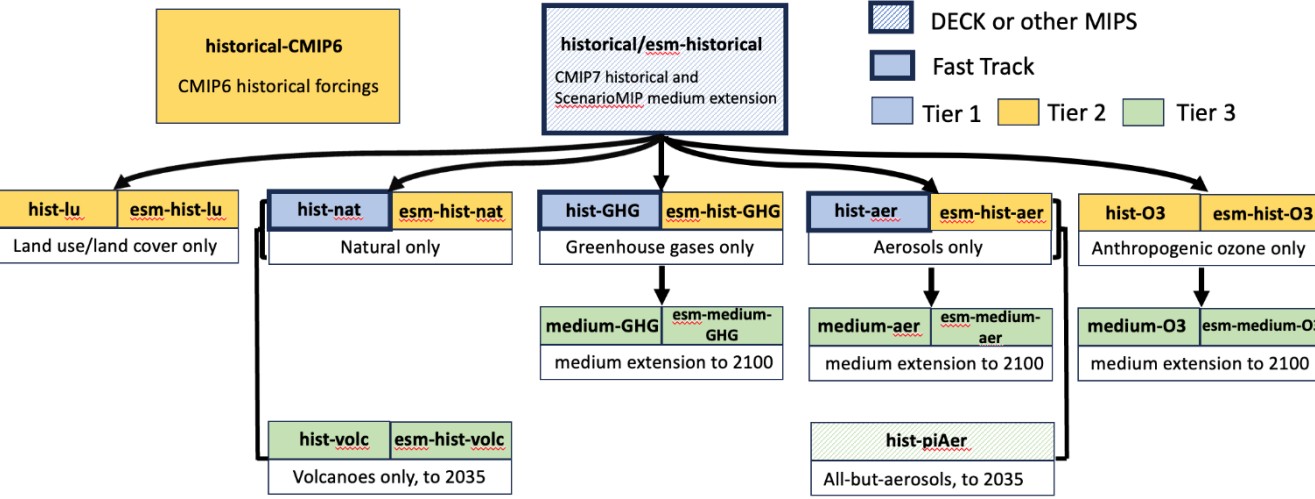

**Figure 2.** Schematic of the experiments proposed under DAMIP v2.0 for CMIP7. Solid arrows indicate the decomposition into separated forced responses. Hatched boxes show simulations which are in other MIPs, but which are of particular relevance to DAMIP.




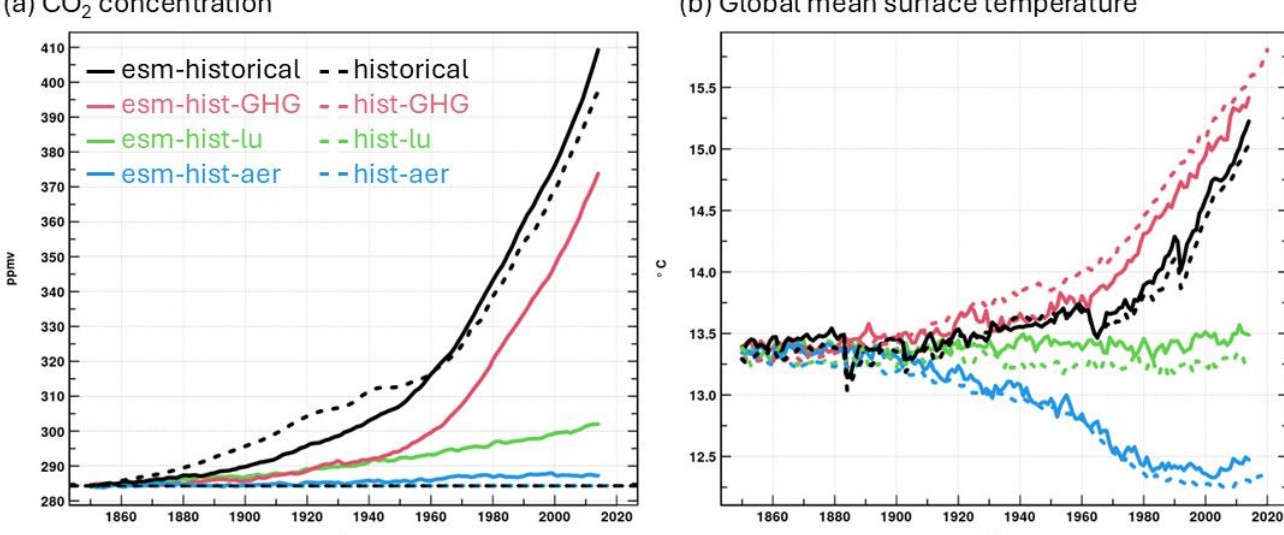

**Figure 3.** Global mean surface $CO_2$ concentration (a) and global mean near-surface air temperature (b) in interactive $CO_2$
experiments performed with CanESM5.0 (Swart et al., 2019). Solid lines show results from interactive $CO_2$ simulations
(esm-historical (black), esm-hist-GHG (red), esm-hist-lu (green), and esm-hist-aer (blue)), and dashed lines show results
from corresponding prescribed $CO_2$ simulations (historical (black), hist-GHG (red), hist-lu (green), and hist-aer (blue)).
Results shown are ensemble means of at least five ensemble members. Note that the $CO_2$ concentration prescribed in hist-
GHG is the same as that in historical, and is fixed at the preindustrial level in hist-lu and hist-aer.