# Peer review of "The Detection and Attribution Model Intercomparison Project"

_EGUsphere, 2024_

## Author Response (AR1)

**Response to review comments on "The Detection and Attribution Model Intercomparison Project (DAMIP v2.0) contribution to CMIP7"**

We would like to thank the reviewers for their detailed and constructive comments on the manuscript. We have made a number of changes to address their comments, and we think that responding to these comments has strengthened the manuscript. Changes to the manuscript in response to each comment are quoted in our responses, and the full set of changes to the manuscript, including other minor editorial changes, can be seen in the tracked changes version of the manuscript included with this submission.

Our responses to the review comments are described in our published responses, a copy of which is included for completeness below. The reviewers' comments are shown in bold, and our responses are shown in plain text.

**AC1**: 'Reply on RC1', Nathan Gillett, 17 Mar 2025

Thanks to the reviewer for the positive review of the manuscript. We have responded to all the requests for changes and clarification of the text. The reviewer's comments are shown in bold below, and our responses are shown in plain text.

**The following are suggested changes:**

**1. Line 83: The word "times" to appear after preindustrial.**

Suggested change made.

**2. Line 132: The concept of additivity should be introduces/explained more as it is not common knowledge.**

An example to explain the meaning of testing additivity has been added "For example, aerosol-only simulations and all-but-aerosol simulations can be used together with simulations including all forcings to test whether the response to aerosols and the response to other forcings add to give the response to all forcings combined (e.g. Simpson et al., 2023)."

**3. Lines 134-137: This should be rephrased for clarity.**

This sentence was complicated by the inclusion of an unnecessary condition concerning additivity. This has now been removed and the text has been modified to improve clarity. The sentence now reads "However, if the objective of an analysis is to characterize the response to one particular forcing, then individual forcing simulations will lead to reduced sampling uncertainties, because they do not require a difference between two sets of simulations to be taken. each of which has its own sampling uncertainties."

**4. Lines 186-189: This should be rephrased for clarity.**

This sentence was separated into two shorter sentences and modified to increase readability and improve clarity. This text now reads "hist-O3 simulations should not be carried out using models with interactive gas-phase chemistry. This is because ozone is simulated interactively in response to changes in ozone depleting substances, methane and other species in such models, and the concentrations of ozone depleting substances, methane and other species do not change in these hist-O3 simulations."

**5. Lines 241-245: Shorter sentences should be used for clarity.**

We have separated this text into shorter sentences and made other changes to improve clarity and readability. This text now reads "While much of the time evolution of biomass burning emissions has occurred as a result of human activity, the historical simulation includes observed year-to-year variations in biomass burning partly driven by natural variability. However, it is not easy to separate human-induced changes in biomass burning from naturally-induced changes. Therefore we request that modelling centres specify constant biomass burning emissions as in the piControl in this hist-nat simulation."

**6. Lines 404-406: Rephrase. Replace "not been completely true" with something along the lines of "there are differences".**

'this has not been completely true' replaced with 'there have been some differences'.

**7. Line 865: The figure 2 does not include any representation of the simulations with Medium scenario forcings up to 2035. The figure should be revisited as it is not obvious what the different boxes are (upper/lower as well as boxes with no connects to other boxes). The yellow boxes could be read in two ways - CMIP6 or Tier2. So better clarity is needed.**

The figure has been revised to state explicitly that the historical simulations are extended to 2035 using the Medium scenario. Text has also been added to caption to clarify this, as well as the meaning of the different boxes:

'Blue boxes are Tier 1 experiments, yellow  boxes are Tier 2 experiments, green boxes are Tier 3 experiments, and white boxes contain descriptive text. All historical simulations except historical-CMIP6 run from 1850 to 2035 using the Medium scenario forcings from 2022, while historical-CMIP6 runs from 1850 to 2035 using the SSP2-4.5 scenario from 2015. The *historical* or *esm-historical* simulation shown in the top row uses CMIP7 historical forcings which can be decomposed into the sets of forcings used in each of the simulations in the second row. Three of these simulations are extended using the Medium scenario from 2036 to 2100 as shown in the third row. The fourth row depicts two additional experiments that are complementary to those in the second row.'

**Citation**: https://doi.org/10.5194/egusphere-2024-4086-AC1

**AC3**: , Nathan Gillett, 24 Apr 2025

**The detection and attribution model intercomparison project is an important part of CMIP6 and data from their experiments have been used in hundreds of studies.**

**I have some comments, that I hope the authors of this important proposal will consider.**

**+ Encouragement of model involvement (Lines 71-73, 498-499)**

**I really like Figure 1 that shows how many models produced DAMIP experiments for CMIP6 and how many studies used them.**

**I estimate that for Chapter 10 in IPCC 2013 they used 44 models with historical simulations, and 16 with historicalNat experiments. But for Chapter 3 in IPCC 2021 the numbers were 59 models with historical, and 14 models with hist-nat experiments. This is a bit disappointing, I hope we can do better for CMIP7.**

**The experiments are extremely useful for understanding the responses within individual models, something institutions should be interested in.**

**Can there be more outreach to encourage more institutions involvement, which will benefit their own model development, as well as be useful for DAMIP.**

Thanks for flagging the take-up of DAMIP v1.0 simulations in hundreds of studies. We agree that it will be important to encourage as many modelling groups as possible to participate in DAMIP 2.0. In 2024 we presented the draft experimental design for DAMIP v2.0 to modelling centre scientists and others at several international conferences including the European Geophysical Union, American Geophysical Union and International Meeting on Statistical Climatology. Of course this manuscript is another required step in describing and publicizing DAMIP to modelling centres. In response to this comment, and as we approach the time over which modelling centres will be running CMIP7 experiments, we will redouble our efforts to publicize DAMIP v2.0 and encourage modelling centres to participate. We also hope that the inclusion of three of the DAMIP experiments in the Assessment Fast Track will ensure that modelling centers participate in those experiments.

**+ Design of hist-nat (Lines 238-247, 252-254)**

**I am surprised to see the design for hist-nat to not include stratospheric ozone changes due to influences of solar and volcanic factors, and put those forcing factors in the hist-O3 with the anthropogenic O3 factors.**

**I think this is a major step backwards. It will no longer be possible to compare anthropogenic only and natural only simulated climate, without adding even more caveats than we do already.**

**I would be reassured if the authors could demonstrate that the natural O3 changes in the stratosphere have ignorable influence on climate in the simulations, but I fear that is not the case for all diagnostics of interest, especially in the stratosphere (Shindell et al, 2013).**

**As it currently stands the design does not "allow the attribution of observed climate change to natural, greenhouse gas and other anthropogenic forcings" (Line 253).**

Thanks for the comment. We did discuss this question amongst the author team, but decided on keeping ozone constant in hist-nat for the following reasons: i) Our complete set of historical forcings simulations consists of hist-nat, hist-GHG, hist-aer, hist-O3 and hist-lu. Natural forcings, well-mixed GHGs, and aerosol and aerosol precursor emissions all affect ozone concentrations. It would be inconsistent to include ozone changes in hist-nat, but not hist-GHG or hist-aer. ii) If we were to include natural ozone changes in hist-nat, and in hist-O3 (as was the case for DAMIP v1.0), this would be a departure from our approach of having each forcing factor appear only once in this set of simulations, and mean that we would no longer be able to assess the additivity of responses by comparing the sum of the responses to hist-nat, hist-GHG, hist-aer, hist-O3 and hist-lu to the response in historical. iii) Requiring natural ozone changes in hist-nat means that a bespoke forcing dataset has to be created for this purpose for DAMIP. This potentially introduces delays and complexity, and in DAMIP v1.0 delayed the start of the hist-nat experiments. Any extension of the hist-nat experiments would similarly require an extension of this forcing dataset. In addition, some modeling centers use their own ozone forcing dataset in their historical simulations, taken from a coupled chemistry simulation with their own model, so we would need to rely on those modeling centers producing their own natural-only forcing dataset in a consistent manner themselves, which would be difficult.

However, these arguments aside, we expect negligible response in tropospheric climate to these ozone changes, and therefore we anticipate that this choice will have negligible influence on DAMIP v2.0 output, with the possible exception of some stratospheric variables.

As suggested by the reviewer, we are beginning a new simulation with CanESM5 of the response to naturally-induced ozone changes only to test this assumption. Results are not yet available at the time of the deadline for revising the manuscript, but if we have another opportunity to make final revisions to the manuscript we will describe the results in the manuscript then. If not, results from this simulation will be described in a future paper reporting the results of DAMIP.

**+ Usefulness of interactive CO2 simulations in D&A (Lines 145-159, 326-345)**

**Could the authors give some examples how interactive CO2 experiments, with emissions of CO2 rather than prescribed CO2, be useful in D&A studies. I see how such experiments can be very interesting and useful in other fields.**

We note in Section 3 of the manuscript "given that direct and accurate observations of the evolution of atmospheric $CO_2$ exist, we recommend that prescribed-$CO_2$ simulations should continue to be used for detection and attribution studies of changes in the physical climate. For this reason, our highest-priority Assessment Fast Track simulations are concentration-driven, as

are the other simulations described in Sections 3.1 and 3.2". However, interactive $CO_2$ experiments could be used for example in D&A studies attributing changes in atmospheric carbon, ocean carbon or land carbon to land use change emissions (esm-hist-lu), fossil $CO_2$ emissions and other well-mixed GHG changes (esm-hist-GHG), and aerosol and aerosol-precursor emissions (esm-hist-aer). Such studies could be carried out using standard detection approaches approaches. We have added a new sentence to Section 3 to address this:

"Such simulations could for example be used to evaluate the effects of biogeochemical feedbacks on the responses to particular forcings, such as aerosols (e.g. Szopa et al., 2021), and could be used in studies attributing changes in atmosphere, land or ocean carbon pools to land use change, fossil $CO_2$ emissions and other GHG changes, and other factors."

More broadly, we think that these interactive $CO_2$ simulations will be helpful for the interpretation of D&A results for physical climate variables based on prescribed-concentration simulations. For example as we now note in the text, even though all D&A analyses to date attribute to changes in concentration, it is generally tacitly assumed that changes attributed to well-mixed GHG concentrations are equivalent to those attributed to well-mixed GHG emissions, while those attributed to aerosols or natural forcings conditional on $CO_2$ being constant, are not strongly influenced by this assumption. For example, if some plausible simulations showed a large $CO_2$ response to historical variations in natural forcings, this would be important information for interpreting temperature changes attributable to natural forcings in standard D&A results. Note that consideration of the effect of carbon cycle feedbacks on the response to natural forcings within DAMIP is similar to consideration of the effects of ozone feedbacks on natural forcings which the reviewer suggests we consider in the previous comment and are investigating. Further, we agree with the comment that these simulations are likely to be useful in fields beyond D&A.

**I have had conversations with several proponents of the experiments, but I remain unconvinced of their use in D&A studies.**

We agree that for attribution of physical climate variables it generally makes sense to continue to use prescribed concentration simulations, as stated in the manuscript.

**A main assumption in D&A is linearity of responses when combining forcings. Has that been demonstrated in interactive CO2 experiments?**

To our knowledge such a set of individual forcing simulations with interactive $CO_2$ has not previously been carried out, so such linear additivity has not been demonstrated. This set of simulations should allow this to be tested in a range of models. Where departures from additivity are found, we agree that these need to be accounted for in any analyses. We note that under such conditions D&A can still be carried out, but it would be important to clarify which component includes any interaction term.

**The example shown in Figure 3 is claimed to show the concentrations of CO2 in CO2 emissions simulations are "relatively close" to that in prescribed concentration simulations. I think that is debatable in the shown model. Also other models give even wider discrepancies between the concentrations in the two set ups (Figure 7 in Sanderson et al, 2024), with equivalently wider surface temperature responses.**

**This would mean adding a radiative forcing uncertainty to one of the few forcing factors with low uncertainties.**

Thanks for the comment. We changed 'relatively close to' to 'within 15 ppm of' to be more quantitative and avoid this subjective judgement. We agree that the agreement is less good for some other models. This is the reason that in Section 3 we write "given that direct and accurate observations of the evolution of atmospheric $CO_2$ exist, we recommend that prescribed-$CO_2$ simulations should continue to be used for detection and attribution studies of changes in the physical climate".

**To avoid confusion, these experiments should be separated from DAMIP.**

As discussed above, these interactive $CO_2$ simulations can be used for D&A of carbon cycle variables, and are relevant to the interpretation of standard D&A results based on prescribed concentration simulations. For these reasons we prefer to keep them as part of DAMIP v2.0. The use of a prefix 'esm-' in the names of the experiments will ensure that users are clear on the distinction between these and the concentration driven simulations. We further note that some modeling centers may only run emissions driven simulations in CMIP7 and if they don't have the opportunity to participate we may end up with fewer modeling centers ultimately participating in DAMIP (a concern raised in the reviewer's first comment).

**+ Extension from 2021 to 2035 (Lines 222, 234-236)**

**Is the proposed extension of 14 years that helpful? This is much longer than we did for CMIP6. Do we have a sense of how much the SSP2-4.5 forcings differed with reality up to present day? That would provide some evidence to support the proposal.**

The extension of the historical simulations to 2035 is already justified in the manuscript where we say "Given that actual anthropogenic forcings are expected to diverge only slightly from the Medium emissions scenario over the first decade or so, we request that these historical simulations and other DAMIP experiments are extended in this way. This will allow researchers to carry out attribution analyses based on contemporary data over the next decade, at least in the absence of a major volcanic eruption, and will likely ensure an overlap with the next phase of CMIP. DAMIP v1.0 simulations were extended in a similar way from 2015 to 2020, but in hindsight this was not long enough, since at the time of writing CMIP7 simulations are not yet available, but observations are available up to the end of 2024, well beyond the end of the DAMIP historical simulations. Such a need is particularly apparent for regularly updated attribution analyses (Forster et al., 2024)". We further note that 2035 is only ten years away now, and by the time the time the DAMIP simulations are completed, it will be rather less than this.

As explained above, DAMIP v1.0 experiments were only extended to 2020, which means that they cannot be used to attribute recent contemporary climate data, such as in the Global Climate Indicators project which provides annually updated estimates of anthropogenic warming (Forster et al., 2024).

We do already discuss how well SSP2-4.5 forcings agree with reality in Section 3.4": "While forcings have evolved broadly consistently with the SSP2-4.5 scenario used to extend DAMIP v1.0 simulations since 2015 (Matthews and Wynes, 2022), there have been some differences. For example, recent aerosol emissions from China have declined more strongly than those specified in the SSP2-4.5 scenario (Wang et al., 2021), a decline in marine sulphur emissions resulting from new International Maritime Organisation regulations was not included in the SSP2-4.5 simulations (Watson-Parris et al., 2022, 2024), the COVID-19 pandemic changed emissions temporarily in ways which weren't included in the SSP2-4.5 scenario (e.g. Szopa et al., 2021), and the Hunga Tonga eruption in 2021 injected sulphur dioxide and water vapour into the stratosphere, influencing the climate (e.g. Jenkins et al., 2023)."

**The CMIP7 forcings task team, are debating about how to update the historical forcings year on year. If they succeed this might mitigate somewhat the requirement to extend to 2035. Has there been any discussion with that team?**

Thanks for the comment. In response to this review comment we contacted the CMIP7 forcings task team, and learned that discussions on how to update the forcings on an annual basis are ongoing. We agree that any such updated forcings could be used to update the simulations and already propose this in the manuscript: "LESFMIP proposed a set of individual forcing simulations using regularly updated forcing estimates, as part of CMIP6 (Smith et al., 2022), though at the time of writing such simulations have not yet been carried out and the protocols and process are not in place for the regular update of forcings which would support such experiments. If this could be achieved in the future, however, such simulations could be used to understand the contribution of updates to particular forcing estimates in contributing to observed climate trends, particularly on interannual to decadal timescales. Given the uncertainty in whether future forcings over the next decade or so will diverge strongly from those specified in the Medium scenario, and if and when updated forcings datasets will be made available to the modelling community, we are not proposing additional named experiments here. However, we do note that the Earth System Grid Federation (ESGF) naming convention includes a forcing index which can be used to label different forcing variants for a given experiment. We recommend that this index is used to publish simulations with updated forcing variants if and when modelling centres perform them." We edited this text to further emphasize the benefits of such updates by adding "as well as to reduce uncertainties in attribution results associated with forcing changes after 2021" after "on interannual to interdecadal timescales".

We suggest that such updated forcing experiments should be in addition to, rather than instead of, an extension of all simulations to 2035. The reason for this is that it is generally easier to carry out a complete set of simulations at one time, and the extra effort needed to download and prepare forcing data published at a later date might mean that fewer modelling centres would run with updated forcings after the CMIP7 Assessment Fast Track. Moreover, in some cases, model versions may be superseded, and it may no longer be straightforward to update simulations with new forcings several years into the future. Finally, the actual number of years involved in the extension of simulations from 2022 to 2035 is small.

**+ Variant ID use suggestion (Lines 201, 424-429)**

**It might be complicated to use the "f" value in the variant ID to indicate specific forcing set ups across all models, in the way suggested. Some institutions have used that to indicate their unique forcing set ups (e.g., HadGEM3, UKESM, GISS-E2, CAMS-CSM1-0) in CMIP6, and the CMIP7 forcings task team have discussed recommending models use the "f" value to be associated with the input forcings version number. Has this proposal been discussed with other MIPs and the CMIP7 forcing team?**

Thanks for flagging this. We contacted the CMIP7 forcing task team, and learned that this issue is still subject to discussion. The task team have considered publishing simulations using updated forcing datasets with the version number of the forcing included in the NetCDF filenames similar to what we are suggesting here, though no decisions have yet been made. Associating the "f" value with the input forcing version number, as the reviewer suggests, is actually similar to what we are proposing here - the only difference is whether a version number or a year are used to designate a particular forcing update. To maintain flexibility we added this sentence to this text "Alternatively a version number of the forcing dataset could be used". A final decision on the most appropriate naming approach should be made once the forcings are available, and would require additional documentation. As we already say: "Any such effort would require additional community coordination and documentation if and when new forcing datasets were published, and would be most valuable if carried out in conjunction with updated all-forcing scenario simulations as part of ScenarioMIP or DCPP".

**+ Additivity/linearity of 5 experiments (Lines 231-232)**

**The recommended hist-GHG, hist-nat, hist-aer, hist-lu and hist-O3 experiments are chosen as they are hoped that summed up they are equal to the historical. It would be helpful to say that with everything else being equal that the variance of the climate response of such a sum would be 5 times greater than the variance of the climate response of historical, due to internal variability. For a multi-model mean this might seem less important, but for examining responses for individual models it could make it more difficult to interpret results.**

As proposed, we have added this sentence "We note that the variance of this sum will in general be five times larger than the variance of the response in historical, which will need to be accounted for when identifying departures from additivity" to Section 3.1.2.

**+ Tuning to the historical record**

**This is a bit of a controversial subject, but it would be helpful to mention the issue that some models have included the historical observed temperature changes - in one way or another - in their model development cycle (Hourdin et al, 2017). Some colleagues have suggested that the DAMIP experiments are then more important, as single forcing experiments are less likely to be impacted by any circular reasoning.**

Thanks for the suggestion. As requested, we have added a sentence mentioning the issue that some modelling centres have used observed temperature changes in model tuning to Section 3.1:

"A larger ensemble of CMIP7 historical simulations will also make it easier to evaluate the consistency of simulated historical climate change with that observed, though any such analysis should account for the use of historical temperature change in the tuning of some models (Hourdin et al. 2017)."

**+ Specific comments:**

**L73-75: I am curious. How are experiments added to DAMIP?**

In each case the experiments were discussed by the DAMIP steering committee in consultation with the CMIP panel, and were described in a peer-reviewed paper. For example, the CovidMIP experiments were proposed in the CovidMIP experimental design paper (Lamboll et al, 2020, https://doi.org/10.5194/gmd-14-3683-2021), and a subset were published as DAMIP simulations after consultation with the DAMIP steering committee.

**L170, L330, L335, and elsewhere: It would be helpful to give section numbers for IPCC references when talking about something very specific. It can be difficult to find what is being referred to.**

We now specify that the reference on line 335 focuses on Figure SPM.2 and we have added section references to the IPCC citations on lines 170 and 335.

**L185-189: I presume the authors mean the experiment name "esm-hist-GHG" and not "hist-GHG" here.**

Thanks for flagging this ambiguity. This sentence was intended to refer to non-$CO_2$ GHGs only, and it applies to both hist-GHG and esm-hist-GHG. The sentence now reads "For hist-GHG and esm-hist-GHG they should specify emissions rather than concentrations of all well-mixed non-$CO_2$ greenhouse gases simulated interactively".

**L234: "… from 1850 to 2035 using …", should be "… from 1850 to 2021 using …"**

Corrected, thanks.

**L280-281: There are difficulties even if not specifying stratospheric ozone in isolation. A prescribed troposphere and stratosphere ozone can still cause issues as the tropopause in the model may not correspond to the input $O_3$, causing unwanted effects (Hardiman et al. (2019)).**

Thanks for pointing us to this paper. While the Hardiman et al. (2019) results are for a rather extreme climate (4xCO2) we now acknowledge this possibility in Section 3.1.2 and caution readers with the following text "Nevertheless, we acknowledge that inconsistencies between tropopause height and ozone concentrations can still exist when prescribing ozone over the full column, which users should be aware could impact conclusions in certain circumstances (Hardiman et al. 2019)".

**L333-335 What does this refer to? All I can find in the IPCC 2021 SPM that this could refer to is figure SPM.2. But that compares the attribution assessment from chapter 3, using concentrations of CO2 (and other forcings) in different models and approaches, with the outputs from simple climate models (chapter 7) driven by different radiative forcing changes,**

**where the CO2 concentrations are constrained to match historic CO2 concentration changes (7.SM.2.2).**

The reviewer is correct – it does refer to Figure SPM.2. This is now clarified. The sentence cited here reads "IPCC (2021, Figure SPM.2) contrasted observationally-constrained attributable warming in response to changes in concentration of greenhouse gases among other factors, with estimates of the warming attributable to emissions of $CO_2$, methane and other species." This is a correct description of the figure. Figure SPM.2c shows attribution to emissions of each of the forcing factors, not to changes in concentration, as noted in the caption. It is based on the analysis shown in Figure 6.12b of Chapter 6 (Szopa et al.,2021).

**L347-350: While I disagree with the hist-nat design, it should be made clear if the esm-hist-nat has the similar no O3 changes from solar and volcanic implemented.**

We've added "including ozone" after "all other forcing held fixed" to make it clear that esm-hist-nat has fixed ozone.

**L419-421 It would be helpful to show how much estimated actual radiative forcings has diverged from ssp2-4.5 over the last decade or so, to give a sense of the expected future uncertainty.**

Matthews and Wynes (2022), whom we cite, showed that $CO_2$ emissions and total non-$CO_2$ forcing evolved similarly to SSP2-4.5 up to 2021 in their Figure 2. We are not aware of a more recent or comprehensive comparison of the actual evolution of forcings with the SSP scenarios, and analysing reported emissions or diagnosing radiative forcings would be beyond the scope of this paper. However, in Section 3.4 we include the following text describing differences between SSP2-4.5 and the evolution of forcings in reality and pointing to relevant references: "While forcings have evolved broadly consistently with the SSP2-4.5 scenario used to extend DAMIP v1.0 simulations since 2015 (Matthews and Wynes, 2022), there have been some differences. For example, recent aerosol emissions from China have declined more strongly than those specified in the SSP2-4.5 scenario (Wang et al., 2021), a decline in marine sulphur emissions resulting from new International Maritime Organisation regulations was not included in the SSP2-4.5 simulations (Watson-Parris et al., 2022, 2024), the COVID-19 pandemic changed emissions temporarily in ways which weren't included in the SSP2-4.5 scenario (e.g. Szopa et al., 2021), and the Hunga Tonga eruption in 2021 injected sulphur dioxide and water vapour into the stratosphere, influencing the climate (e.g. Jenkins et al., 2023)."

In response to this review comment, we have added a reference forward to Section 4 after the text referred to in this comment.

**Figure 2 (page 30) Tiers (1 to 3) are shown here, but "Tiers" for the CMIP7 are not mentioned in the main text.**

Thanks for flagging this. We added this sentence to the start of Section 3 "As in DAMIP v1.0, we designate the highest priority experiments as Tier 1 experiments, with Tier 2 and Tier 3 representing successively lower priority experiments (see Figure 2)", which points to Figure 2 for the details of which tiers each experiment is assigned to. We also edited a sentence in 3.1.2 to identify the Tier 1 experiments: "This set includes the hist-nat, hist-GHG and hist-aer experiments which are

designated as Assessment Fast Track experiments here (Dunne et al., 2024) and also as Tier 1 experiments".

**Citation**: https://doi.org/10.5194/egusphere-2024-4086-AC3

**AC2**: 'Reply on RC2', Nathan Gillett, 23 Apr 2025

**One of the biggest criticisms of the CMIP project is that there are too many subprojects and a tendency to be "all things for all people". Given human and hardware resource limitations, this puts tremendous pressure and stress on the climate modeling community. That being said, the DAMIP and SCENARIOMIP projects are among the most important components of the CMIP due to their relevance to policy and decision makers.**

**The paper is well written and explains the DAMIP project thoroughly. I do, however, have some strong opinions and I will use this venue to express them. My hope is that the authors will consider them.**

**First, the three Fast Track Tier 1 experiments, natural forcings only, anthropogenic well-mixed greenhouse gases only, and anthropogenic aerosols only are far more important than the Tier 2 and 3 experiments and modeling groups should strongly consider devoting their limited resources to the Tier 1 experiments.**

**Second, the Large Ensemble Single Forcing Model Intercomparison Project (LESFMIP) offers many new analysis opportunities. This is particularly true for attribution studies, where simple statistical comparisons of histograms from the different experiments can be insightful. Linear regression attribution algorithms have always been difficult to explain to those not directly involved in attribution while histogram comparisons offer simpler visual explanations and can make the case more strongly for the human influence, if any, on the climate system topics being considered. I recommend that the paper more strongly endorse large ensembles, especially for the Tier 1 experiments.**

Thanks very much for the positive comments on the manuscript. We note the concern regarding overstretching the modelling community, and we agree that for groups with limited resources it makes sense to focus on the three Assessment Fast Track experiments. That said, we think that there is value in the other experiments we are proposing, so we have retained them in DAMIP v2.0.

We expect that the designation of hist-nat, hist-GHG and hist-aer as Assessment Fast Track experiments will mean that modelling groups will prioritise them and likely focus more resources on them. We did already have a sentence encouraging groups to run larger ensembles where possible "Note that we also request an ensemble size of at least three for all other DAMIP v2.0 historical simulations, though we encourage groups to run larger ensembles if possible". To further emphasize these points, and to link to LESFMIP, we now follow this existing sentence "This set includes the hist-nat, hist-GHG and hist-aer experiments which are designated as Assessment Fast Track experiments here (Dunne et al., 2024), based on the extensive use of the corresponding DAMIP v1.0 simulations in the literature and in support of IPCC assessment reports" with this one "For this reason, we ask modelling groups to prioritize these simulations, and to consider running large ensembles of these simulations to support extreme event attribution and other applications (e.g. Smith et al., 2022), if time and resources allow."

**Third, no connection to HighResMIP2 is made in the paper. It has become quite clear that simulation of most types of extreme weather event requires far higher resolution than the standard resolution of CMIP6. While the HighResMIP1 organizers were focused on tropical**

**cyclone, the improvement in simulation of extreme weather events is not limited to this one storm type. At horizontal resolutions of ~25km the simulated gradients of both temperature and moisture are far sharper and more realistic than at 100km or coarser. This is a critical feature in simulating intense storms and revealing super Clausius-Clapeyron scaling, if any, in changes in extreme precipitation. Better resolution of mountains also aids in simulating more realistic heatwaves and even blocking events can be improved in some regions.**

Thanks for flagging the link to HighResMIP2, which we agree is very relevant for event attribution of many extreme events. We have added a new paragraph to Section 4 to describe links with HighResMIP2:

"High resolution climate model simulations are often able to better represent phenomena relevant to climate extremes, such as tropical cyclones, atmospheric rivers, and heat waves and heavy rainfall in mountainous regions (e.g. Roberts et al., 2025). Such events are often the focus of extreme event attribution studies. The HighResMIP2 (Roberts et al., 2025) 1950s control (control-1950), and historical simulation beginning in 1950 (hist-1950), as well as its extension to 2100 with the Medium scenario will be valuable in supporting extreme event attribution for such variables, with allowance made for the fact that the 1950s control is not a preindustrial control. We also encourage modelling groups to carry out DAMIP experiments with high resolution models to the extent possible, but realize that computational constraints may prevent this. Finally, we note that the HighResMIP2 choice of the Medium scenario to extend historical simulations to 2100 aligns with our choice of the same scenario in DAMIP v2.0."

**Citation**: https://doi.org/10.5194/egusphere-2024-4086-AC2

---

## Author Response (AR2)

**Description of two minor modifications to the accepted version of "The Detection and Attribution Model Intercomparison Project (DAMIP v2.0) contribution to CMIP7"**

We are pleased that our manuscript has been accepted for publication in GMD. In preparing the final publication-ready manuscript, we made two changes to the text to better respond to issues raised by the reviewers in the light of information newly available since our paper was accepted.

Gareth Jones review included the following comment (available here EGUsphere - The Detection and Attribution Model Intercomparison Project (DAMIP v2.0) contribution to CMIP7, along with our response):

**It might be complicated to use the "f" value in the variant ID to indicate specific forcing set ups across all models, in the way suggested. Some institutions have used that to indicate their unique forcing set ups (e.g., HadGEM3, UKESM, GISS-E2, CAMS-CSM1-0) in CMIP6, and the CMIP7 forcings task team have discussed recommending models use the "f" value to be associated with the input forcings version number. Has this proposal been discussed with other MIPs and the CMIP7 forcing team?**

As indicated in our response we contacted the CMIP7 forcings task team to ask about this, and received a preliminary response. After submission of our updated manuscript, we received additional feedback from the task team recommending that we do not suggest using the "f" value to denote simulations run with updated forcings. Based on this additional feedback, in Section 3.4 we replaced:

*However, we do note that the Earth System Grid Federation (ESGF) naming convention includes a forcing index which can be used to label different forcing variants for a given experiment. We recommend that this index is used to publish simulations with updated forcing variants if and when modelling centres perform them. For example, if the original version of hist-nat, using the Medium scenario forcings from 2022 to 2035 were published with the 'f1' forcing index, and were a major volcanic eruption to occur in 2027, and an updated stratospheric aerosol forcing datasets be published that year, a new version of the simulation using the updated forcings could be published with the 'f2027' forcing index. Such a simulation could cover only part of the time period covered by the original simulation (e.g. 2022-2035 only), as long as it is labelled appropriately. We suggest labelling such simulations with major updates to forcings with the year which historical forcings were extended to e.g. 'f2027'. Alternatively a version number of the forcing dataset could be used.*

with

*However, the CMIP community is continuing to explore how forcings can be updated more regularly and how such updates could be used in CMIP7. DAMIP will engage with this discussion to ensure that developed solutions support updates to attribution simulations.*

In addition, Gareth Jones included this comment in his review:

**I am surprised to see the design for hist-nat to not include stratospheric ozone changes due to influences of solar and volcanic factors, and put those forcing factors in the hist-O3 with the anthropogenic O3 factors. I think this is a major step backwards. It will no longer be possible to compare anthropogenic only and natural only simulated climate, without adding even more**

**caveats than we do already. I would be reassured if the authors could demonstrate that the natural O3 changes in the stratosphere have ignorable influence on climate in the simulations, but I fear that is not the case for all diagnostics of interest, especially in the stratosphere (Shindell et al, 2013).**

As part of our response (EGUsphere - The Detection and Attribution Model Intercomparison Project (DAMIP v2.0) contribution to CMIP7), we wrote the following:

*As suggested by the reviewer, we are beginning a new simulation with CanESM5 of the response to naturally-induced ozone changes only to test this assumption. Results are not yet available at the time of the deadline for revising the manuscript, but if we have another opportunity to make final revisions to the manuscript we will describe the results in the manuscript then. If not, results from this simulation will be described in a future paper reporting the results of DAMIP.*

This simulation has now been carried out, and based on its results we added two additional sentences to the description of hist-nat in Section 3.1.2 to describe the results and better address the reviewer's concern:

*To test the sensitivity to this change in experimental design, we carried out a test simulation with CanESM5.0 of the response to the DAMIP v1.0 specified solar and volcanically-induced ozone changes alone. This showed small forced changes in global mean stratospheric temperature of less than 0.5°C, but no discernible changes in tropospheric climate.*